# The Role of Circulating Biomarkers in Patients with Coronary Microvascular Disease

**DOI:** 10.3390/biom15020177

**Published:** 2025-01-25

**Authors:** Rossella Quarta, Giovanni Martino, Letizia Rosa Romano, Giovanni Lopes, Francesco Fabio Greco, Carmen Anna Maria Spaccarotella, Ciro Indolfi, Antonio Curcio, Alberto Polimeni

**Affiliations:** 1Department of Pharmacy, Health and Nutritional Sciences, University of Calabria, 87036 Rende, Italy; 2Division of Cardiology, Annunziata Hospital, 87100 Cosenza, Italy; 3Department of Medical and Surgical Sciences, Magna Graecia University, 88100 Catanzaro, Italy; 4Division of Interventional Cardiology, Annunziata Hospital, 87100 Cosenza, Italy; 5Division of Cardiology, Department of Advanced Biomedical Sciences, Federico II University, 80134 Naples, Italy

**Keywords:** circulating biomarkers, coronary microvascular dysfunction (CMD), coronary flow reserve (CFR), index of microvascular resistance (IMR)

## Abstract

Coronary microvascular disease (CMD) comprises a spectrum of conditions characterized by the functional and structural abnormalities of coronary microcirculation, affecting vessels typically smaller than 500 μm. Despite its clinical significance as a contributor to myocardial ischemia, CMD frequently remains underdiagnosed due to the limitations of current diagnostic approaches. Invasive testing, including coronary reactivity assessment, is considered the gold standard, but it is resource-intensive and not always accessible. Non-invasive methods, such as positron emission tomography (PET) and transthoracic Doppler echocardiography (TTDE), offer alternatives but are limited by varying accuracy and accessibility. Amid these diagnostic challenges, there is increasing interest in circulating biomarkers as adjuncts in CMD evaluation. Biomarkers associated with endothelial dysfunction, inflammation, and oxidative stress, detectable through routine blood tests, may assist in CMD diagnosis, risk stratification, and therapeutic monitoring. These biomarkers can offer insights into CMD pathogenesis and enable early, non-invasive screening to identify patients who may benefit from more invasive investigations. This narrative review examines studies assessing biomarkers in CMD patients with diagnoses confirmed through invasive techniques. Our objective is to focus on circulating biomarkers linked to the invasive evaluation of coronary microcirculation, aiming to advance the understanding of the underlying mechanisms of this prevalent condition and enhance diagnostic accuracy and the clinical management of affected patients.

## 1. Introduction

Coronary microvascular dysfunction (CMD), which affects vessels usually smaller than 500 μm in diameter, such as pre-arterioles, arterioles, and capillaries, is becoming more widely acknowledged as a serious pathological condition marked by structural and functional abnormalities in coronary microcirculation [1,2]. These tiny vessels are crucial regulators of coronary blood flow, accounting for about 70% of coronary resistance. Inward arteriolar remodeling, capillary rarefaction, intravascular plugging, perivascular fibrosis or infiltration, and extramural compression (e.g., myocardial hypertrophy, increased left ventricular end-diastolic pressure [LVEDP]) are structural changes in the coronary microcirculation [3]. Meanwhile, functional changes involve impaired vasodilator response and increased vasoconstrictor activity, such as microvascular spasms, which primarily result from endothelial and smooth muscle cell dysfunction [3].

CMD is a key contributor to myocardial ischemia and presents clinically as various conditions, including angina with non-obstructive coronary artery disease (ANOCA), ischemia with non-obstructive coronary artery disease (INOCA) [4], and myocardial infarction with non-obstructive coronary artery disease (MINOCA) [5,6,7]. In addition, CMD plays an important role in other cardiac and systemic diseases, such as hypertrophic cardiomyopathy, systemic autoimmune diseases, diabetes mellitus, and heart failure with preserved ejection fraction (HFpEF) [2,8,9,10].

To enhance the understanding of CMD, a clinical–pathogenetic classification is proposed, dividing CMD into four main categories: (1) CMD occurring in the absence of myocardial diseases or obstructive coronary artery disease (CAD), (2) CMD associated with myocardial diseases, (3) CMD related to obstructive epicardial CAD, and (4) iatrogenic CMD [11]. This classification underscores the diverse etiologies of CMD and highlights the need for tailored diagnostic and therapeutic approaches [12].

The pathogenesis of CMD is complex and multifactorial, involving mechanisms such as endothelial dysfunction, inflammation [13], oxidative stress, and altered vasoreactivity [14]. These processes collectively contribute to impaired coronary flow reserve, microvascular spasms, and increased coronary microvascular resistance, leading to the clinical manifestations of CMD [7]. Given the heterogeneity of its mechanisms, CMD is prevalent in up to 50% of patients with chronic coronary syndromes and over 20% of those presenting with acute coronary syndrome (ACS) [9].

Despite its considerable impact on patient outcomes, CMD remains underdiagnosed primarily due to challenges associated with its detection. Invasive coronary reactivity testing, including the assessment of coronary physiology, remains the gold standard for diagnosing CMD [15,16]. Due to its limited availability and the potential risks associated with invasive testing, alternative noninvasive techniques such as positron emission tomography (PET), transthoracic Doppler echocardiography (TTDE), stress echocardiography [17], and cardiac magnetic resonance imaging (CMR) have been employed, although their effectiveness can vary [18] (Figure 1).

Given these diagnostic challenges, there is a growing interest in identifying circulating biomarkers that could enhance the diagnosis, risk assessment, and management of patients with CMD. Biomarkers related to key processes like endothelial dysfunction, inflammation, and oxidative stress offer promising avenues for improving CMD detection and treatment strategies [19].

This narrative review aims to provide a comprehensive analysis of the role of circulating biomarkers in patients with CMD. Specifically, we focus on key biomarkers involved in fundamental pathological processes such as endothelial dysfunction, inflammation, and oxidative stress. A targeted search was performed using major databases, including PubMed and Scopus, to identify studies evaluating circulating biomarkers in patients diagnosed with CMD through invasive methods, such as coronary flow reserve (CFR) and the index of microcirculatory resistance (IMR). The novelty of this review lies in its emphasis on studies employing these advanced techniques, ensuring high diagnostic accuracy and providing deeper insights into the underlying pathophysiological mechanisms of CMD.

Therefore, for each pathophysiological mechanism involved in CMD, we prioritize studies in which CMD was evaluated invasively (Figure 2). This approach ensures an analysis based on the current gold standard, allowing for differentiation between the endothelium-dependent and endothelium-independent forms of CMD.

## 2. Pathophysiology of Coronary Microvascular Disease

The proximal coronary compartment consists of epicardial coronary arteries, which are conductance vessels with a cross-sectional diameter greater than 500 μm. These vessels, visible on coronary angiography, are the primary sites for obstructive atherosclerosis. Under normal conditions, the epicardial arteries account for less than 10% of coronary vascular resistance and only become hemodynamically significant when more than 70% of the arterial lumen is narrowed. The distal compartment includes coronary microcirculation, encompassing all vessels with a cross-sectional diameter smaller than 500 μm. This includes pre-arteriolar vessels (100–500 μm in diameter), intramural arterioles (diameter less than 100 μm), and capillaries [1,2].

CMD is characterized by a multifaceted interplay of functional and structural disruptions within the coronary microcirculation, resulting in impaired coronary blood flow (CBF) and myocardial ischemia, even in the absence of significant epicardial coronary artery stenosis. The pathophysiology of CMD can generally be divided into functional and structural abnormalities, each of which plays a crucial role in reducing coronary flow reserve and contributing to the clinical manifestations of the disease [14,15] (Figure 1).

### 2.1. Functional Alterations

The functional disturbances inherent to CMD primarily involve the dysregulation of vasomotor tone, presenting as impaired vasodilation and the enhanced vasoconstriction of coronary microvasculature [14,20]. Endothelial dysfunction is central to these functional impairments. By producing vasodilatory mediators such as prostacyclin, nitric oxide (NO), and endothelium-derived hyperpolarizing factors (EDHFs), the endothelium controls vascular tone under normal physiological conditions [1,2]. However, in CMD, there is a marked reduction in the bioavailability of these mediators, particularly NO, largely due to increased oxidative stress. Reactive oxygen species (ROS) play a crucial role in this process by degrading NO, thereby diminishing its vasodilatory effect and simultaneously enhancing the vasoconstrictive actions of endothelin-1 (ET-1). This imbalance contributes to impaired vasodilation and increased microvascular resistance.

Endothelium-independent mechanisms, though less well understood, are thought to involve the impaired relaxation of vascular smooth muscle cells (VSMCs), the increased sensitivity of VSMCs to typical vasoconstrictor stimuli, and abnormal autonomic regulation. Additionally, ET-1 may play a role in these processes [11,21].

Inflammatory processes involved in CMD further exacerbate functional dysregulation. Inflammatory mediators such as interleukin-6 (IL-6), tumor necrosis factor-alpha (TNF-α), and C-reactive protein (CRP) promote endothelial activation and reduce NO bioavailability, aggravating the impaired vasomotor response. These inflammatory pathways, combined with oxidative stress-induced NO degradation, result in a diminished ability of the microvasculature to respond adequately to increased myocardial oxygen demand, leading to ischemic symptoms even in the absence of epicardial coronary artery stenosis [21,22].

In this context, diabetes mellitus (DM) significantly contributes to CMD primarily through oxidative stress, and it is characterized by reduced NO bioavailability, increased reactive oxygen species (ROS), elevated endothelin, and systemic inflammation. CMD often precedes hyperglycemia in type 2 diabetes, highlighting the role of insulin resistance and inflammation in its progression. Diabetes-associated CMD manifests systemically, affecting multiple organs, including the heart, retina, and kidneys [9].

Furthermore, coronary microvascular tone, regulated by myogenic mechanisms, metabolic control, endothelial function, circulating factors, and autonomic innervation, is significantly influenced by adrenergic innervation during exercise or the presence of endothelial dysfunction while contributing minimally at rest in healthy coronary circulations. Adrenergic receptors (α and β) modulate coronary circulation, with β2 receptors primarily mediating vasodilation, whereas α-receptor stimulation induces vasoconstriction, except for α2 receptors on endothelial cells, which promote vasodilation [9].

Autonomic dysregulation significantly contributes to CMD pathophysiology. An imbalance between sympathetic and parasympathetic activity exacerbates coronary vasoconstriction, particularly in clinical scenarios such as diabetes mellitus, hypertension, and post-myocardial infarction, where intrinsic vasodilatory mechanisms are already compromised. This autonomic imbalance, coupled with the increased vasoconstrictive actions of ET-1 and reduced vasodilatory capacity, further impairs coronary microvascular function and exacerbates myocardial ischemia. Overall, the interplay between oxidative stress, inflammation, and autonomic dysfunction creates a vicious cycle of impaired coronary vasoreactivity, contributing to the functional alterations seen in CMD [1].

### 2.2. Structural Alterations

The structural remodeling of coronary microvasculature is another critical aspect of CMD. This remodeling includes hypertrophic changes in small coronary arteries and arterioles, characterized by luminal narrowing and increased medial wall thickening due to smooth muscle hypertrophy and enhanced collagen deposition. Perivascular fibrosis and capillary rarefaction, manifested as a reduction in capillary density, are also prominent features. These structural alterations are particularly severe in conditions associated with increased left ventricular mass, such as hypertrophic cardiomyopathy and hypertensive heart disease, where they significantly impair coronary microvascular function [2]. In both primary and secondary left ventricular hypertrophy (LVH), CMD is marked by capillary rarefaction and the adverse remodeling of intramural arterioles, leading to increased wall-to-lumen ratios and progressive ischemic damage. This contributes to fibrosis, ventricular remodeling, and heightened risks of systolic dysfunction and heart failure. In HFpEF, CMD exacerbates oxygen demand mismatches and myocardial injury, even without obstructive coronary disease, driven by endothelial dysfunction, systemic inflammation, and oxidative stress, especially in patients with cardiometabolic risk factors [23].

The molecular mechanisms underlying these structural changes are complex, involving a cascade of pathological processes driven by oxidative stress and chronic inflammation. Key signaling pathways, such as the RhoA/Rho-kinase pathway, are implicated in promoting vascular smooth muscle contraction, pathological vascular remodeling, and perpetuating endothelial dysfunction [24]. Furthermore, epigenetic mechanisms, such as DNA methylation and histone modifications, further regulate vascular inflammation and aging, offering the potential for more precise diagnostics and management [25].

Given the complex and multifactorial nature of CMD, there is a growing interest in the potential role of circulating biomarkers as tools for the early detection, risk stratification, and management of this condition. Circulating biomarkers could provide insights into the underlying pathophysiological processes, such as endothelial dysfunction, oxidative stress, and inflammation, which are central to CMD [14]. By reflecting the biochemical milieu associated with CMD, these biomarkers could serve as non-invasive indicators of disease activity, helping identify patients at risk, monitor disease progression, and potentially guide therapeutic interventions. Indeed, the identification of specific biomarkers linked to key molecular pathways, such as NO bioavailability or ROS production, could facilitate the development of targeted therapies aimed at mitigating the functional and structural impairments seen in CMD. In this context, circulating biomarkers hold promise as valuable adjuncts in the comprehensive assessment and management of patients with CMD.

## 3. Diagnostic Approaches for Coronary Microvascular Disease

The Coronary Vasomotion Disorders International Study Group (COVADIS) established standardized international criteria for diagnosing CMD. These criteria include the presence of clinical symptoms, the absence of obstructive epicardial coronary artery disease, and evidence of myocardial ischemia, ascertained through both non-invasive tests and invasive assessments of coronary microvascular impairment [26,27,28,29,30,31].

In this context, both invasive and non-invasive diagnostic techniques have been developed to assess the functional integrity of the coronary microvasculature and to rule out obstructive coronary artery disease (CAD) as the underlying cause of symptoms. Despite significant advances in diagnostic methods, diagnosing CMD remains challenging.

Invasive methods, such as coronary reactivity testing, are currently considered the most accurate and reliable approach for diagnosing CMD and are regarded as the gold standard due to their capacity to provide detailed insights into microvascular function.

### 3.1. Invasive Diagnostic Techniques

Invasive testing is considered the gold standard for diagnosing CMD, especially in patients with a high clinical suspicion of microvascular dysfunction after obstructive CAD has been excluded. Different tools that are Doppler- and thermodilution-based are available for assessing microvascular function. A comprehensive evaluation of coronary microvascular function encompasses both endothelium-independent and endothelium-dependent assessments [30,32].

Endothelium-independent microvascular function: This is primarily assessed using CFR, the index of microvascular resistance (IMR), and hyperaemic microvascular resistance (hMR). CFR is calculated as the ratio of maximal coronary blood flow during pharmacologically induced hyperemia (commonly with adenosine) to resting flow, reflecting the combined vasodilatory capacity of both epicardial and microvascular coronary arteries. IMR, on the other hand, specifically evaluates microvascular resistance. Using thermodilution techniques, it is calculated as the product of distal coronary pressure and the mean transit time during hyperemia, providing a precise measure independent of epicardial artery function. The pathological values of CFR below 2.0–2.5 or an IMR greater than 25 indicate CMD [4]. hMR is another Doppler-based valuable index, calculated as the ratio of distal coronary pressure to average peak velocity during hyperemia, with values greater than 2.5 indicating CMD [33].

Additionally, continuous thermodilution techniques allow for the quantification of true coronary blood flow (Q) and microvascular resistance (Rµ) without the need for hyperemic agents like adenosine. These techniques are considered to have lower variability compared to traditional methods and provide a more accurate assessment of microvascular function.

Coronary vasospasm testing: The diagnosis of coronary microvascular vasospasm, which may contribute to CMD, involves provocative testing with intracoronary acetylcholine (ACh). This test evaluates endothelium-dependent microvascular function by inducing vasoconstriction or spasms in the microvasculature. A positive test, defined by the reproduction of angina symptoms and ischemic electrocardiographic changes in the absence of significant epicardial spasms, confirms microvascular spasms as contributors to CMD.

Coronary angiography: While coronary angiography is primarily used to exclude obstructive CAD, it can also provide indirect evidence of CMD. For instance, the “coronary slow-flow phenomenon”, characterized by the delayed opacification of coronary vasculature, suggests increased microvascular resistance and/or microvascular spasm—a hallmark feature of CMD [2,14,15].

### 3.2. Non-Invasive Diagnostic Techniques

Non-invasive diagnostic techniques are increasingly utilized in the evaluation of CMD, particularly in patients for whom invasive testing is not feasible. These methods generally evaluate the surrogate markers of coronary microvascular function and include modalities such as TTDE, PET, and CMR.

TTDE measures coronary flow velocity reserve (CFVR), analogous to CFR, by assessing maximal diastolic flow in the left anterior descending artery at rest and during pharmacologic stress. TTDE is advantageous due to its low cost, lack of radiation exposure, and widespread availability. However, it is technically challenging and requires significant operator expertise.

PET is considered the reference standard for the non-invasive assessment of myocardial blood flow (MBF) and CFR. It allows for a comprehensive evaluation of all coronary territories simultaneously by quantifying MBF at rest and during maximal hyperemia. Nevertheless, PET’s use is limited by high costs, limited availability, and radiation exposure.

CMR offers high-resolution imaging capable of assessing myocardial perfusion and distinguishing between epicardial and microvascular dysfunction. CMR stress perfusion, typically performed with adenosine, allows for the quantification of myocardial perfusion reserve (MPR) and has shown promising results in detecting CMD. Additionally, late gadolinium enhancement on CMR has prognostic value, as it is associated with a higher risk of adverse cardiovascular events.

Dynamic Myocardial Perfusion Computed Tomography (CT) is an emerging technique that combines both the anatomical and functional assessments of coronary arteries. When combined with CTCA for the exclusion of obstructive CAD, dynamic myocardial perfusion CT offers promising diagnostic capabilities, though at the cost of increased radiation exposure [1,2,14].

## 4. Circulating Biomarkers in Coronary Microvascular Dysfunction

In the study of CMD, the identification of reliable biomarkers is crucial for understanding the underlying pathophysiological mechanisms and improving diagnosis and management. A comprehensive understanding of endothelial markers, particularly those associated with inflammation and oxidative stress, provides valuable insights into the pathophysiology of CMD. The key indicators of endothelial dysfunction, such as the von Willebrand factor (VWF), angiotensin-converting enzyme (ACE), and soluble adhesion molecules including sVCAM-1, sICAM-1, and E-selectin, not only reflect vascular injury but also underline the inflammatory and oxidative processes contributing to microvascular impairment [34].

Building on this framework, other biomarkers have also garnered attention due to their potential roles in CMD. Among these, the brain-derived neurotrophic factor (BDNF) has been studied for its critical function in maintaining vascular homeostasis and endothelial integrity [35]. Beyond these effects, BDNF has been implicated in modulating adverse cardiac remodeling [36] and promoting inflammation and atherosclerosis [37].

Additionally, heat shock proteins (HSPs), such as HSP60 and HSP72, have emerged as important mediators of microvascular and myocardial damage, particularly in patients with idiopathic left ventricular dysfunction [38]. Notably, HSP60 plays a pivotal role in the pathogenesis of cardiovascular diseases through its multifaceted involvement in mitochondrial function, protein folding, cardiac remodeling and failure, cardiomyocyte regulation, the promotion of atherosclerosis, and the activation of inflammatory pathways [39].

Although numerous biomarkers have been analyzed in the literature, this section focuses on biomarkers that have been studied specifically in patients where CMD was assessed using invasive diagnostic techniques, ensuring a high level of accuracy in evaluating the role of these biomarkers (Table 1).

### 4.1. Traditional Biomarkers of Heart Injury

To date, the role of troponin (Tn) as a biomarker in CMD remains not fully established. Several studies have explored its potential significance in CMD. For instance, research by Takashio et al. showed elevated plasma troponin T (TnT) levels in individuals with CMD compared to healthy controls [40]. Similarly, Fujii et al. observed that patients undergoing elective percutaneous coronary angioplasty (PTCA) exhibited higher post-PTCA values of IMR in those with elevated troponin I (TnI), pointing to significant microvascular dysfunction and evaluating the potential protective effect of statins [41].

In line with this, another study confirmed a similar association between post-PTCA values and plasma creatine kinase MB (CK-MB) levels [42], while a different study involving 55 PTCA patients demonstrated a link between post-PTCA CFR values and cTnT and CK levels [43].

Conversely, a study of 19 patients found no significant correlation between cTnI levels and CFR and a moderate association with IMR [44].

### 4.2. Biomarkers of Inflammation

The role of inflammation in CMD is increasingly recognized as a central mechanism in the pathogenesis and progression of the disease. Elevated levels of circulating biomarkers such as CRP have been consistently associated with microvascular dysfunction in various cardiovascular conditions [45]. CRP, a well-known inflammatory biomarker, is elevated in patients with cardiac syndrome X (CSX) [46,47] and has been shown to serve as a marker of disease activity, suggesting that inflammation plays a significant role in CMD pathophysiology [2,48].

Only a few studies have correlated circulating biomarkers with objective evidence of microvascular dysfunction using invasive tests. For instance, Ong et al. conducted ACh testing in patients with angina and unobstructed coronary arteries to evaluate coronary spasms. Notably, elevated levels of high-sensitivity CRP (hs-CRP) and soluble CD40 ligand (sCD40L) were significantly associated with coronary spasm, hence indicating an inflammatory component in CMD [49]. Furthermore, Teragawa et al. also used intracoronary ACh infusion to assess endothelial function and found that elevated CRP levels were associated with reduced coronary blood flow response [50].

Additional studies have examined the relationship between hs-CRP levels and coronary vasospastic angina [51], as well as the association between inflammation and microvascular resistance in patients with CSX. Using CFR and the IMR, Long et al. demonstrated that elevated hs-CRP levels were linked to impaired CFR and higher IMR, further supporting the role of inflammation in CMD [52].

### 4.3. Biomarkers of Oxidative Stress

Oxidative stress, defined as an imbalance between the production of reactive oxygen species (ROS) and the body’s antioxidant defenses, is a major contributor to cardiovascular disease [53], CSX [54], plaque instability [55], and INOCA [56]. This imbalance leads to reduced NO bioavailability and promotes vascular damage.

An important antioxidant, glutathione, has been linked to CMD, with lower levels indicating greater oxidative stress and impaired microvascular function. Studies have demonstrated that glutathione is an independent predictor of CMD, reinforcing its potential as a biomarker of oxidative stress in this condition. Notably, the only study to evaluate coronary microvascular function invasively found that lower glutathione levels independently predicted impaired microvascular function and increased plaque necrotic core [57].

### 4.4. Biomarkers of Endothelial Dysfunction

Endothelial dysfunction is a key feature of CMD, characterized by impaired NO synthesis and increased vasoconstriction.

ET-1, the most potent endogenous vasoconstrictor, is a critical biomarker of endothelial dysfunction. James Theuerle et al. investigated the relationship between plasma levels of ET-1 and adrenomedullin (ADM) and their association with CMD and cardiovascular outcomes. The study found that elevated ET-1 levels were linked to higher microvascular resistance, while higher levels of ADM were associated with improved coronary flow and a reduced risk of major adverse cardiovascular events (MACEs) [58].

Moreover, lower circulating MOTS-c levels are associated with impaired coronary endothelial function in patients without significant coronary lesions [59].

### 4.5. Miscellaneous Biomarkers

Several other biomarkers have been studied in relation to CMD, even though they do not fit neatly into the aforementioned categories.

Homocysteine, a sulfur-containing amino acid, has been linked to CMD, with elevated levels correlating with impaired microvascular function [60].

Mangiacapra et al. found a significant association between elevated cholesterol levels, particularly LDL cholesterol, and coronary microvascular dysfunction [61].

Pacheco et al. found that higher cBIN1 levels were linked to CMD, suggesting a potential progression from CMD to HFpEF. These findings indicate that cardiomyocyte dysfunction may play a role in the pathophysiology of CMD in INOCA [62].

Soluble urokinase-type plasminogen activator receptor (suPAR) is another potential biomarker that has been associated with coronary microvascular dysfunction. In patients with non-obstructive coronary artery disease, plasma suPAR levels were found to be an independent predictor of coronary microvascular function [63].

Moreover, thrombogenicity has been associated with CMD in patients with acute myocardial infarction (AMI) [64].

Girum Mekonnen et al. from the NHLBI-sponsored Women’s Ischemia Syndrome Evaluation—Coronary Vascular Dysfunction (WISE-CVD) study explored the relationship between CMD and circulating progenitor cells (CPCs) in women with ischemic symptoms but without obstructive coronary artery disease (CAD). In this study, CMD was evaluated through CFR using intracoronary adenosine. The results revealed that a lower CFR, indicating impaired microvascular function, was associated with higher levels of CPCs (CD34+, CD34+/CD133+, and CD34+/CXCR4+), suggesting that chronic ischemia in CMD may stimulate CPC mobilization [65].

Muroya et al. explored the relationship between omega-3 polyunsaturated fatty acids (PUFAs) and CMD in patients with stable coronary artery disease. The study assessed coronary microvascular function using the hMRI and found that lower levels of the eicosapentaenoic acid (EPA)/arachidonic acid (AA) ratio were associated with higher hMRI, indicating worse CMD [66].

Akhiyat et al. found that patients with CMD had significantly lower circulating levels of α-Klotho, a protein involved in vascular health and aging. The reduced α-Klotho levels were associated with CMD [67].

Serotonergic neurons in the central nervous system and enterochromaffin cells in the gastrointestinal tract are the primary producers of serotonin, also known as 5-hydroxytryptamine (5-HT), which platelets from the bloodstream subsequently collect [68]. It has been determined that plasma serotonin levels may serve as a biomarker for CMD in patients with nonobstructive coronary arteries and probable angina. Odaka et al. included 198 individuals who performed an acetylcholine provocation test to evaluate CMD—which is defined as myocardial lactate production without obvious epicardial spasms. A serotonin concentration above 9.55 nmol/L is a powerful independent predictor of the presence of CMD according to the findings, which also indicate a substantial correlation between elevated plasma serotonin levels and CMD [69].

Other studies explored the role of microRNAs (miRNAs) in endothelial and coronary dysfunction. MicroRNAs (miRNAs) are short, highly conserved RNA molecules that regulate gene expression through mRNA silencing, either by transcript degradation or translational repression [70]. These investigations aim to identify miRNA markers associated with both macrovascular and microvascular dysfunction, potentially offering new insights into the diagnosis and management of CMD [71,72]. Notably, the increased expression of miR-224-5p in circulating extracellular vesicles is associated with reduced CFR, indicating endothelial dysfunction. This miRNA was found to originate predominantly from hepatic cells, and it is linked to a higher expression of endothelial ICAM-1, which may contribute to vascular inflammation [73].

Even high levels of uric acid are associated with impaired coronary microvascular function, making it a relevant mostly gender-related marker in the study of CMD [74]. A study by Megha Prasad demonstrated that elevated uric acid levels are associated with inflammation and CMD in postmenopausal women. The research highlighted that higher serum uric acid was correlated with impaired endothelial function, as measured by acetylcholine-mediated coronary blood flow, and increased inflammatory markers, including hs-CRP and neutrophil counts. Uric acid levels have been found to be elevated in women with CMD, suggesting a potential gender-specific biomarker for CMD [75].

According to a pilot study by Keeley et al., women with coronary microvascular dysfunction have lower levels of omega-3 fatty acid-derived mediators, maresin 1 and resolvin D1, suggesting that the pathophysiology of CMD may be influenced by the impaired production of these mediators [76]. Furthermore, Ito et al. in patients without obstructive coronary artery disease (CAD), found a significant correlation between CMD and circulating malondialdehyde-modified low-density lipoprotein (MDA-LDL) levels [77].

**Table 1 biomolecules-15-00177-t001:** Circulating biomarkers and invasive measures of coronary microcirculation.

Authors	Year	Biomarkers Evaluated	N. Subjects	Main Findings	Invasive Measures
Takashio et al. [40]	2013	cTnT, BNP, hs-CRP, IL-6, TNF-α	90 patients 47 controls	cTnT levels were higher in patients with CMD and increased LVEDP	CFR
Kitabata et al. [42]	2013	CK-MB	24 patients	CK-MB levels were higher in patients with a higher MVRI	MVRI
Herrmann et al. [43]	2001	CK, cTnT	55 patients	rCVR after successful coronary stenting was independently associated with elevation of cTnT and CK	CFVR
Park et al. [44]	2014	cTnI	19 patients	cTnI was correlated with IMR but not with CFR values	CFR, IMR
Ong et al. [49]	2015	sCD40L, hs-CRP	62 patients	Patients with Ach-induced coronary spasm and no significant coronary stenoses had elevated levels of sCD40L and hs-CRP	ACh provocation test
Teragawa et al. [50]	2004	CRP	46 patients	Increased CRP levels had reduced CBF response to acetylcholine	ACh provocation test
Hung et al. [51]	2005	hs-CRP	428 patients	hs-CRP was associated with coronary vasospastic angina	Ergonovine provocation test
Long et al. [52]	2017	hs-CRP	20 patients 20 controls	hs-CRP was positively correlated with IMR	CFR, IMR
Dhawan et al. [57]	2011	Cystine/Glutathione, hs-CRP	47 patients	Lower plasma glutathione levels and higher cystine/glutathione ratios were associated with impaired coronary microvascular function and greater plaque necrotic core content	CFVR, hMR, IVUS
Theuerle et al. [58]	2019	ET-1, ADM	32 patients	Elevated ET-1 levels were associated with increased IMR. ADM levels were correlated with CFR.	IMR, CFR
Qin et al. [59]	2018	MOTS-c	20 patients 20 controls	MOTS-c levels were positively correlated with both microvascular and epicardial endothelial function	ACh provocation test
Ahmad et al. [60]	2020	Homocysteine	1418 patients	Elevated homocysteine levels positively correlate with CMD and early atherosclerosis	ACh provocation test
Mangiacapra et al. [61]	2012	Total cholesterol, LDL-c, HDL-c, TG	95 patients	Significant correlation between IMR and total cholesterol and LDL-c	IMR
Pacheco et al. [62]	2021	cBIN1	95 patients 50 controls	Higher cBIN1 score was associated with vasoconstriction to Ach	CFR, ACh provocation test
Mekonnen et al. [63]	2015	suPAR	66 patients	Higher suPAR levels correlated with lower CFR	CFR
Kang et al. [64]	2021	P-FCS, D-dimer	116 patients	P-FCS significantly increased the risk of CMD post-procedurally	IMR
Mekonnen et al. [65]	2016	CPCs	123 patients	Lower CFR was associated with higher levels of CPCs cells	CFR
Muroya et al. [66]	2018	EPA/AA, DGLA	108 patients	Lower levels of the EPA/AA were significantly correlated with increased hMVRI	hMVRI
Akhiyat et al. [67]	2024	α-Klotho	98 patients	Patients with CMD had significantly lower circulating α-Klotho levels	CFR
Odaka et al. [69]	2016	Serotonin	198 patients	Plasma serotonin concentration was found to be significantly higher in patients with CMD	ACh provocation test
James et al. [73]	2022	miR-224-5p	120 patients	miR-224-5p was found to be significantly elevated in the low-CFR group	CFR
Kuwahata et al. [74]	2010	UA	194 patients	Elevated UA levels were associated with impaired CBF	ACh and papaverin provocation test
Prasad et al. [75]	2017	UA	229 patients	Elevated UA levels were associated with impaired CBF	ACh provocation test
Keeley et al. [76]	2022	Resolvins, Maresin 1, EPA, DHA and 18-HEPE	31 patients 12 controls	Women with CMD had significantly lower levels of resolvin D1 and maresin 1 but higher levels of DHA and 18-HEPE compared to the control group	CFR
Ito et al. [77]	2024	MDA-LDL	95 patients	Elevated levels of MDA-LDL were significantly associated with CMD in patients without CAD	CFR, IMR

**Abbreviations:** 18-HEPE, 18-hydroxy-eicosapentaenoic acid; ACh, acetylcholine; ADM, adrenomedullin; BNP, B-type natriuretic peptide; CAD, coronary artery disease; CBF, coronary blood flow; cBIN1, cardiac bridging integrator 1; CFR, coronary flow reserve; CFVR, coronary flow velocity reserve; CK, creatine kinase; CK-MB, creatine kinase myocardial band; CMD, coronary microvascular dysfunction; CPCs, circulating progenitor cells; CRP, C-reactive protein; cystine/glutathione, ratio of oxidized cystine to reduced glutathione; cTnI, cardiac troponin I; cTnT, cardiac troponin T; DGLA, dihomo-gamma-linolenic acid; DHA, docosahexaenoic acid; D-dimer, fibrin degradation product; EPA, eicosapentaenoic acid; EPA/AA, eicosapentaenoic acid/arachidonic acid ratio; ET-1, endothelin-1; hMVRI, hyperemic microvascular resistance index; hs-CRP, high-sensitivity C-reactive protein; HDL-c, high-density lipoprotein cholesterol; IL-6, interleukin-6; IMR, index of microcirculatory resistance; LDL-c, low-density lipoprotein cholesterol; LVEDP, left ventricular end-diastolic pressure; MDA-LDL, malondialdehyde-modified low-density lipoprotein; miR, microRNA; MOTS-c, mitochondrial open reading frame of 12S rRNA-C; MVRI, microvascular resistance index; P-FCS, platelet-fibrin clot strength; rCVR, relative coronary vascular resistance; sCD40L, soluble CD40 ligand; suPAR, soluble urokinase-type plasminogen activator receptor; TG, triglycerides; TNF-α, tumor necrosis factor alpha; total cholesterol, total cholesterol; UA, uric acid; α-Klotho, alpha-klotho.

## 5. Therapeutic Implications of Biomarkers in Coronary Microvascular Disease

The management of CMD remains a significant clinical challenge due to the lack of large-scale randomized trials confirming effective treatment strategies. Current approaches emphasize addressing cardiovascular risk factors such as smoking, hypertension, diabetes, and obesity while tailoring therapies to individual phenotypic presentations. Animal models, from rodents to large mammals, are essential for exploring CMD’s role in ischemia and metabolic disorders. While small models facilitate genetic studies, large models like swine better mimic human physiology, enabling the evaluation of targeted therapies under controlled conditions and ensuring translational relevance to clinical applications [78].

Inflammation, as indicated by elevated CRP and hs-CRP levels, underscores the importance of anti-inflammatory strategies. Statins, beyond their cholesterol-lowering effects, reduce vascular inflammation, enhance nitric oxide bioavailability, and improve endothelial function. These benefits have been observed not only in patients with obstructive coronary artery disease but also in those with CMD and non-obstructive CAD [79,80]. Similarly, angiotensin-converting enzyme (ACE) inhibitors and angiotensin II receptor blockers (ARBs) exert vasoprotective effects by improving microcirculatory function, reducing oxidative stress, and enhancing CFR [81,82].

Beta-blockers remain highly effective in reducing chest pain episodes in CMD by lowering myocardial oxygen demands and inducing endothelium-dependent vasodilation. Third-generation beta-blockers, such as nebivolol, offer additional benefits by improving endothelial function and exercise tolerance [83].

Nitrates, particularly short-acting formulations, provide symptomatic relief in CMD by inducing vasodilation [84]. Emerging therapies, such as phosphodiesterase type-5 inhibitors (e.g., sildenafil), have demonstrated potential in improving microvascular function by enhancing nitric oxide signaling [85]. Similarly, the inhibition of Rho-kinase pathways with agents like fasudil shows promise in preventing microvascular spasms and improving myocardial perfusion [86,87].

Ranolazine, widely used in CMD, reduces myocardial oxygen demand and alleviates ischemic symptoms [88], while other anti-anginal agents, including ivabradine, nicorandil, and trimetazidine, also target oxygen consumption and have demonstrated benefits in alleviating angina symptoms [89]. Additional promising therapeutic roles may be played by agents such as SGLT2 inhibitors, GLP-1 receptor agonists, and novel lipid-lowering therapies, for which specific studies are still needed.

Additionally, endothelial dysfunction, as evidenced by biomarkers like ADM and ET-1, underscores the therapeutic potential of interventions aimed at improving endothelial function. Notably, Zibotentan, a selective endothelin-A receptor antagonist, has emerged as a promising therapeutic agent targeting ET-1. By inhibiting ET-1 activity, Zibotentan may reduce vascular tone and improve microvascular function, offering a novel approach to managing CMD and its associated complications [90].

The integration of biomarkers into CMD management not only advances the understanding of underlying mechanisms but also enables a more precise and individualized therapeutic approach. Combining traditional therapies with emerging pharmacological strategies and lifestyle modifications offers a comprehensive framework for improving outcomes in CMD patients. Ongoing research into innovative treatments, including those targeting inflammation, endothelial dysfunction, and oxidative stress, will be pivotal in addressing unmet clinical needs in CMD management.

## 6. Conclusions

CMD represents a complex and multifactorial condition with significant implications for cardiovascular health. As this review has highlighted, CMD is underpinned by intricate interactions between functional and structural abnormalities within the coronary microcirculation, driven by endothelial dysfunction, inflammation, oxidative stress, and autonomic dysregulation. The challenges in diagnosing CMD, particularly given its subtle and non-obstructive nature, underscore the importance of invasive diagnostic methods such as coronary reactivity testing, which remains the gold standard. However, the advent of circulating biomarkers offers a promising adjunct to these invasive techniques, providing insights into the underlying pathophysiology and opening avenues for more personalized therapeutic strategies. By targeting specific biomarkers associated with inflammation, oxidative stress, and endothelial dysfunction, novel therapies can be developed to mitigate the impact of CMD, ultimately improving patient outcomes.

## Figures and Tables

**Figure 1 biomolecules-15-00177-f001:**
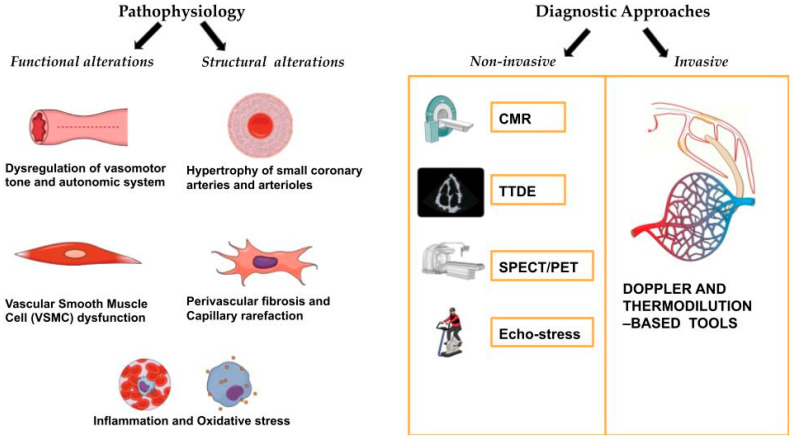
Pathophysiology and diagnostic approaches to CMD.

**Figure 2 biomolecules-15-00177-f002:**
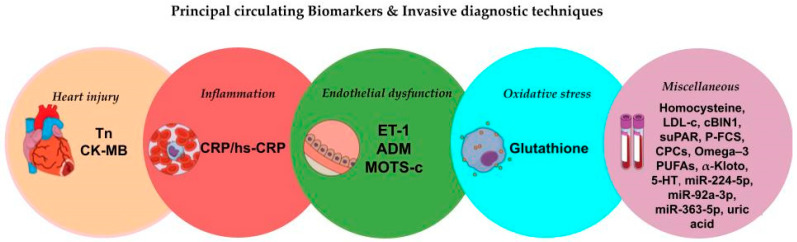
Principal circulating biomarkers evaluated with invasive diagnostic techniques.

## Data Availability

Not applicable.

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
