# Peer review of "The Role of Circulating Biomarkers in Patients with Coronary Microvascular Disease"

_biomolecules, 2025, doi:10.3390/biom15020177_

Round 1
Reviewer 1 Report
Comments and Suggestions for Authors
The present narrative review is interesting. However, coronary microvascular dysfunction (CMD) encompasses several pathogenetic mechanisms involving coronary microcirculation. It plays a significant role in determining myocardial ischemia in patients with angina without obstructive coronary artery disease and several other conditions, including nonischemic cardiomyopathies and heart failure, especially the phenotype associated with preserved ejection fraction.
Notably, as coronary microvascular dysfunction has been strongly associated with the development of heart failure with preserved ejection fraction and cardiac disease in diabetic patients, further research is needed.
The authors must include a paragraph on the pathogenetic role of cardiovascular autonomic dysfunction in CMD and possible biomarkers. This addition will significantly enhance the comprehensiveness of the review.
Furthermore, a more extensive comment on the role of Diabetes Mellitus is warranted. In DM, oxidative stress and other biomarkers are more pronounced.
Overall, the data are interesting. The author's work is valuable to cardiology, providing a deeper understanding of the complex interplay of factors in coronary microvascular dysfunction.
Author Response
The present narrative review is interesting. However, coronary microvascular dysfunction (CMD) encompasses several pathogenetic mechanisms involving coronary microcirculation. It plays a significant role in determining myocardial ischemia in patients with angina without obstructive coronary artery disease and several other conditions, including nonischemic cardiomyopathies and heart failure, especially the phenotype associated with preserved ejection fraction. Notably, as coronary microvascular dysfunction has been strongly associated with the development of heart failure with preserved ejection fraction and cardiac disease in diabetic patients, further research is needed. The authors must include a paragraph on the pathogenetic role of cardiovascular autonomic dysfunction in CMD and possible biomarkers. This addition will significantly enhance the comprehensiveness of the review. Furthermore, a more extensive comment on the role of Diabetes Mellitus is warranted. In DM, oxidative stress and other biomarkers are more pronounced. Overall, the data are interesting. The author's work is valuable to cardiology, providing a deeper understanding of the complex interplay of factors in coronary microvascular dysfunction.
We appreciate the reviewer’s insightful comment on the importance of cardiovascular autonomic dysfunction in the pathogenesis of CMD. Recognizing its significant role, we have expanded this section in our review. Additionally, we have integrated a more extensive discussion on Diabetes Mellitus, emphasizing its role in driving inflammation and oxidative stress, which are key contributors to the development of CMD (Pag. 5 142-147, 148-155).
Reviewer 2 Report
Comments and Suggestions for Authors
The theme of this review is interesting in general. However, I have some comments:
1) It is not clear from the Introduction what is the scientific novelty of the review presented? In the Introduction (before the objective), clear reference to other reviews on similar topics should be given and the novelty of the topic of the current review should be explained.
2) Describe in detail the methods of review.
3) It is advisable to summarise the main points of the review, which have the character of scientific novelty and/or synthesis of scientific information, in a table with an indication of sources.
4) Some promising markers (which associated with endothelial and/or microcirculatory damage and may also be relevant for CMD) are not covered in this review:
- Brain-derived neurotrophic factor (see DOIs: 10.1111/j.1582-4934.2012.01621.x, 10.15275/rusomj.2022.0202, 10.1038/s41598-019-51776-8, etc)
- Heat shock proteins (see DOIs: 10.1379/CSC-272.1, 10.3389/fmolb.2020.00073, etc)
- Monocyte chemotactic protein-1 (see DOIs: 10.3389/fcvm.2024.1280734, etc) and other markers of inflammation.
5) The therapeutic implications can certainly be greatly expanded.
Author Response
The theme of this review is interesting in general. However, I have some comments:
1) It is not clear from the Introduction what is the scientific novelty of the review presented? In the Introduction (before the objective), clear reference to other reviews on similar topics should be given and the novelty of the topic of the current review should be explained.
We appreciate this observation and have clarified the scientific novelty of our review in the revised Introduction. Specifically, our review focuses on studies investigating the correlation between circulating biomarkers and invasive assessment of the coronary microcirculation, which remains the gold standard for CMD evaluation. (Pag.3 86-92)
2) Describe in detail the methods of review.
As this is a narrative review rather than a systematic review, we did not adhere to the methodologies typically associated with systematic literature reviews. However, we conducted a targeted literature search using two primary databases, PubMed and Scopus, and reviewed the references of previously published studies to identify relevant articles. Our selection criteria were centered on studies that met two essential conditions: the evaluation of circulating biomarkers and a diagnosis of CMD confirmed through invasive techniques. (Pag. 3 86-92)
3) It is advisable to summarise the main points of the review, which have the character of scientific novelty and/or synthesis of scientific information, in a table with an indication of sources.
Considering our specific focus on the correlation between circulating biomarkers and invasive evaluation of the coronary microcirculation, we have included a table summarizing the main studies addressing this topic as requested. This table also provides the corresponding references for clarity and completeness.
4) Some promising markers (which associated with endothelial and/or microcirculatory damage and may also be relevant for CMD) are not covered in this review:
- Brain-derived neurotrophic factor (see DOIs: 10.1111/j.1582-4934.2012.01621.x, 10.15275/rusomj.2022.0202, 10.1038/s41598-019-51776-8, etc)
- Heat shock proteins (see DOIs: 10.1379/CSC-272.1, 10.3389/fmolb.2020.00073, etc)
- Monocyte chemotactic protein-1 (see DOIs: 10.3389/fcvm.2024.1280734, etc) and other markers of inflammation.
We acknowledge the relevance of these biomarkers and have introduced them into our review (Pag. 8 291-303). However, a detailed discussion of these markers falls outside the primary scope of our work due to the lack of data correlating these biomarkers with invasive evaluation of the coronary microcirculation.
5) The therapeutic implications can certainly be greatly expanded.
​​As suggested, we have expanded the section on therapeutic implications, providing a more comprehensive overview of potential treatment strategies for CMD (Pag.12 495-527, 534-541).
Reviewer 3 Report
Comments and Suggestions for Authors
Authors focused on clinically neglected impact of circulating biomarkers in patients with coronary microvascular disease that is underdiagnosed due to the limitations of current diagnostic approaches. While biomarkers associated with inflammation and oxidative stress which promote endothelial dysfunction might be detectable through routine blood tests. Thereby, may assist in diagnosis, risk stratification, and therapeutic monitoring of patients with coronary microvascular disease. This narrative review includes articles assessing biomarkers in patients with coronary microvascular disease diagnosed through invasive techniques. This approach may help to understand the relevant mechanisms implicated in microvascular disease and consequently to enhance diagnostic accuracy and the clinical
management.
According to the idea of authors by targeting specific biomarkers associated with inflammation, oxidative stress, and endothelial dysfunction, novel therapies can be developed to mitigate the impact of coronary microvascular disease, ultimately improving patient outcomes.
However, specific biomarkers of inflammation and oxidative stress contribute not only to endothelial and microvascular dysfunction but also to the dysfunction of the cardiomyocytes (f.e. in left ventricular hypertrophy, ischemia or heart failure, Camici et al 2020)
What authors think about in the context of their idea?
Recently, experimental animal models of coronary microvascular dysfunction were reviewed (Sorop et al. 2020) as important contributors to myocardial ischemia and metabolic disorders. It provides models for testing novel therapeutic interventions.
What authors think about in the context of their review?
Moreover, what authors think about epigenetic mechanisms involved in microvascular disease and diagnostic testing strategy? See f.e. Masi et al. 2021.
Finally, endothelial cell markers are comprehensively described in older paper Goncharov et al. 2017 that may be included in the list of specific biomarkers of coronary microvascular diseases. What authors think about?
It would be appreciated to increase font the font in Figure 1 and 2.
Author Response
Authors focused on clinically neglected impact of circulating biomarkers in patients with coronary microvascular disease that is underdiagnosed due to the limitations of current diagnostic approaches. While biomarkers associated with inflammation and oxidative stress which promote endothelial dysfunction might be detectable through routine blood tests. Thereby, may assist in diagnosis, risk stratification, and therapeutic monitoring of patients with coronary microvascular disease. This narrative review includes articles assessing biomarkers in patients with coronary microvascular disease diagnosed through invasive techniques. This approach may help to understand the relevant mechanisms implicated in microvascular disease and consequently to enhance diagnostic accuracy and the clinical management.
According to the idea of authors by targeting specific biomarkers associated with inflammation, oxidative stress, and endothelial dysfunction, novel therapies can be developed to mitigate the impact of coronary microvascular disease, ultimately improving patient outcomes.
However, specific biomarkers of inflammation and oxidative stress contribute not only to endothelial and microvascular dysfunction but also to the dysfunction of the cardiomyocytes (f.e. in left ventricular hypertrophy, ischemia or heart failure, Camici et al 2020)
What authors think about in the context of their idea?
We appreciate the reviewer’s insightful observation regarding the impact of inflammation and oxidative stress on cardiomyocyte dysfunction. Biomarkers associated with these processes not only contribute to endothelial and microvascular impairment but also play a crucial role in the progression of cardiomyocyte injury. This interplay is particularly evident in conditions such as left ventricular hypertrophy, ischemia, and heart failure, where the disruption of microvascular and cardiomyocyte function creates a pathological feedback loop. Recognizing the importance of this relationship, we have integrated this concept into our review (Pag. 5 175-182)
Recently, experimental animal models of coronary microvascular dysfunction were reviewed (Sorop et al. 2020) as important contributors to myocardial ischemia and metabolic disorders. It provides models for testing novel therapeutic interventions.
What authors think about in the context of their review?
We recognize the role of experimental animal models in advancing our understanding of CMD and its contribution to myocardial ischemia and metabolic disorders. These models, when carefully selected and designed, offer valuable insights into the mechanisms underlying CMD and provide a platform for testing novel therapeutic interventions. However, it is essential to consider the inherent limitations and translational challenges when applying findings from animal studies to clinical practice. We have discussed this issue in the text (Pag 12 499-503).
Moreover, what authors think about epigenetic mechanisms involved in microvascular disease and diagnostic testing strategy? See f.e. Masi et al. 2021.
The importance of epigenetic mechanisms in the pathogenesis of CMD is undeniable, as they represent a critical layer of regulation influencing inflammation, oxidative stress, and endothelial function. Mechanisms such as DNA methylation and histone modifications are pivotal for understanding the complex disease process. Following the reviewer’s suggestion, we have incorporated a brief discussion on this topic into our review to highlight its relevance. (Pag. 6 187-190)
Finally, endothelial cell markers are comprehensively described in older paper Goncharov et al. 2017 that may be included in the list of specific biomarkers of coronary microvascular diseases. What authors think about?
The review by Goncharov et al. (2017) is indeed comprehensive and provides valuable insights into endothelial markers. In our review, we have primarily focused on studies that highlight the correlation between circulating biomarkers and invasive assessments of the coronary microcirculation, which remain the gold standard for CMD evaluation. Nonetheless, we acknowledge the importance of their work and its contribution to the broader understanding of endothelial dysfunction in CMD, and we have cited it in our review. (Pag. 8 284-290)
It would be appreciated to increase font the font in Figure 1 and 2.
We appreciate this suggestion and have revised Figures 1 and 2 to ensure improved readability by increasing the font size.
Reviewer 4 Report
Comments and Suggestions for Authors
The article offer a comprehensive review of biomarkers role in coronary microvascular disease. The pathophysiology of the disease is presented and the diagnosis methods explained. Markers of inflammation, oxidative stress and endothelial dysfunction are of particular interest, proved in subjects with invasively confirmed diagnosis of microvascular dysfunction. Authors bring an interesting approach of suggesting potential therapeutic novel approaches to microvascular dysfunction.
Author Response
The article offer a comprehensive review of biomarkers role in coronary microvascular disease. The pathophysiology of the disease is presented and the diagnosis methods explained. Markers of inflammation, oxidative stress and endothelial dysfunction are of particular interest, proved in subjects with invasively confirmed diagnosis of microvascular dysfunction. Authors bring an interesting approach of suggesting potential therapeutic novel approaches to microvascular dysfunction.
We thank the reviewer for their positive feedback on our work.
Round 2
Reviewer 2 Report
Comments and Suggestions for Authors
I approve current version of paper.